# A pulse-chasable reporter processing assay for mammalian autophagic flux with HaloTag

**Willa Wen-You Yim[1†], Hayashi Yamamoto[1,2*†], Noboru Mizushima[1*]**

[1]Department of Biochemistry and Molecular Biology, Graduate School of Medicine, The University of Tokyo, Tokyo, Japan; [2]Department of Molecular Oncology, Institute for Advanced Medical Sciences, Nippon Medical School, Tokyo, Japan

**Abstract** Monitoring autophagic flux is necessary for most autophagy studies. The autophagic flux assays currently available for mammalian cells are generally complicated and do not yield highly quantitative results. Yeast autophagic flux is routinely monitored with the green fluorescence protein (GFP)-based processing assay, whereby the amount of GFP proteolytically released from GFP-containing reporters (e.g. GFP-Atg8), detected by immunoblotting, reflects autophagic flux. However, this simple and effective assay is typically inapplicable to mammalian cells because GFP is efficiently degraded in lysosomes while the more proteolytically resistant red fluorescent protein (RFP) accumulates in lysosomes under basal conditions. Here, we report a HaloTag (Halo)-based reporter processing assay to monitor mammalian autophagic flux. We found that Halo is sensitive to lysosomal proteolysis but becomes resistant upon ligand binding. When delivered into lysosomes by autophagy, pulse-labeled Halo-based reporters (e.g. Halo-LC3 and Halo-GFP) are proteolytically processed to generate Halo[ligand] when delivered into lysosomes by autophagy. Hence, the amount of free Halo[ligand] detected by immunoblotting or in-gel fluorescence imaging reflects autophagic flux. We demonstrate the applications of this assay by monitoring the autophagy pathways, macroautophagy, selective autophagy, and even bulk nonselective autophagy. With the Halo-based processing assay, mammalian autophagic flux and lysosome-mediated degradation can be monitored easily and precisely.

## Editor's evaluation

This paper describes the development of quantitative approaches to measure the turnover of cellular material in the lysosome. These new tools are carefully validated and the data shows their application to several physiologically relevant types of autophagy. This assay will be very useful for any researcher interested in studying autophagy flux

## Introduction

Autophagy is the lysosome-based degradation system in eukaryotes essential for cellular homeostasis as part of intracellular quality control and intracellular remodeling during environmental adaptation (*Bento et al., 2016*; *Kawabata and Yoshimori, 2020*; *Noda and Inagaki, 2015*; *Zhao et al., 2021*). Dysregulated autophagy is associated with aging and diseases (*Fleming et al., 2022*; *Kawabata and Yoshimori, 2020*; *Klionsky et al., 2021a*; *Mizushima and Levine, 2020*; *Zhao et al., 2021*). Macroautophagy and microautophagy are major autophagy pathways by which cytoplasmic material is delivered to lysosomes for degradation. The former pathway involves the formation of an autophagosome that sequesters material and fuses with lysosomes while the latter pathway occurs via

**\*For correspondence:**
hayashi-yamamoto@nms.ac.jp
(HY);
nmizu@m.u-tokyo.ac.jp (NM)

[†]These authors contributed equally to this work

direct invagination of the lysosomal membrane (*Kawabata and Yoshimori, 2020*; *Noda and Inagaki, 2015*; *Zhao et al., 2021*). 'Autophagy' will refer to macroautophagy and microautophagy, collectively hereafter. Autophagic activity is the uninterrupted series of events that begins with material entering lysosomes and ends with their degradation by lysosomal hydrolases. The amount of autophagic degradation is often referred to as autophagic flux (*Klionsky et al., 2021b*).

Monitoring autophagic flux is integral to studying autophagy (*Ohsumi, 2014*). Quantitative assays have been developed for *Saccharomyces cerevisiae*, with the GFP-Atg8 processing assay being the most commonly used (*Shintani and Klionsky, 2004*). GFP-Atg8 localizes to both outer and inner autophagosomal membranes and is delivered into the vacuole as part of autophagic bodies (inner membrane-delimited vesicles), where GFP-Atg8 is then processed by vacuolar proteases. Unlike Atg8, GFP is relatively stable against vacuolar proteolysis and accumulates in the vacuole as autophagy progresses. The amount of free GFP detected by immunoblotting thus reflects autophagic flux. Additionally, this assay can be adapted to monitor specific forms of autophagy by replacing Atg8 in the reporter with one's protein of interest, such as with the mitochondrial protein Om45 to monitor mitophagy (selective autophagy of mitochondria) (*Kanki and Klionsky, 2008*) or the cytosolic protein Pgk1 to monitor bulk nonselective autophagy (*Welter et al., 2010*). The versatility and ease of implementing the reporter processing assay are reasons why it is routinely used in yeast autophagy studies.

Implementing the GFP-Atg8 processing assay in mammalian cells (using LC3 and GABARAP family proteins) has been challenging because GFP is rapidly degraded in mammalian lysosomes (*Katayama et al., 2008*). Only a few studies have succeeded in detecting the release of free GFP (*Gao et al., 2008*; *Hosokawa et al., 2006*). Although the success rate could be increased by raising the luminal pH of lysosomes with nonsaturating concentrations of lysosomotropic agents (*Gao et al., 2008*; *Klionsky et al., 2021b*), doing so might also affect GFP-LC3 processing itself and thus would require careful optimization. The difficulties in properly executing the GFP-LC3 processing assay prompted researchers to develop other assays.

In mammalian cells, autophagic flux is often estimated from the amount of selective autophagic cargo degraded or, with the LC3 turnover assay, the amount of LC3-II produced in the presence of a lysosomal inhibitor (i.e. the amount of LC3-II that would have otherwise been degraded; *Klionsky et al., 2021b*; *Mizushima and Murphy, 2020*; *Mizushima and Yoshimori, 2007*). Autophagic flux can also be monitored with the fluorescent reporters, RFP-GFP-LC3 (*Kimura et al., 2007*) and GFP-LC3-RFP (*Kaizuka et al., 2016*), with which autophagic flux is determined from the reduction in GFP fluorescence compared to the relatively unchanging fluorescence of the lysosome-resistant RFP and cytosol-localizing RFP, respectively. However, each assay has its own limitations (for a full list, see *Klionsky et al., 2021b*; *Mizushima and Murphy, 2020*). Some require the use of a lysosomal inhibitor control, which can limit the assay's dynamic range (*Liebl et al., 2022*) and cause side-effects (*Florey et al., 2015*; *Juhász, 2012*), while others are complicated or are low in sensitivity. Failing to consider and address the limitations may lead to misinterpretations of autophagic flux.

Here, we report the HaloTag (Halo)-based processing assay, a simple and effective method to quantitatively monitor autophagic flux in mammalian cells. We found that Halo becomes proteolytically resistant upon ligand binding. Pulse-labeling with Halo ligand creates a pool of Halo-based reporters that, when delivered into lysosomes by autophagy, are proteolytically processed to release stable free Halo[ligand] (Halo covalently conjugated with its ligand). The amount of free Halo[ligand] thus reflects autophagic flux specifically. Moreover, Halo-based reporters can also be used by fluorescence microscopy and other means. We demonstrate the applications of this assay by monitoring macroautophagy, selective autophagy, and bulk nonselective autophagy.

## Results

### A HaloTag-LC3B processing assay to quantify autophagic flux in mammalian cells

Although a few studies have succeeded in doing so (*Gao et al., 2008*; *Hosokawa et al., 2006*), implementing the equivalent of the routinely used yeast GFP-Atg8 processing assay (*Klionsky et al., 2021b*) in mammalian cells is challenging. With monomeric GFP (mGFP)-LC3B, a small amount of free mGFP was produced in wild-type HeLa cells but not in *FIP200* knockout (KO) cells under both growing and starvation conditions, indicating that free mGFP was generated in an autophagy-dependent manner

(*Figure 1a*). However, although a starvation-induced decrease of mGFP-LC3B levels was detected, an increase in free mGFP levels was not (*Figure 1a*, left) as mGFP is degraded in mammalian lysosomes like other GFPs (*Katayama et al., 2008*). Replacing mGFP with the proteolytically resistant monomeric RFP (mRFP) (*Katayama et al., 2008*) allowed for an increase in free mRFP levels upon starvation to be detected (*Figure 1a*, right), but the increase was not apparent due to the pre-existing mRFP band (*Figure 1a*, right) that probably resulted from basal autophagy-mediated lysosomal accumulation (*Katayama et al., 2008*). Hence, it is difficult to detect mammalian autophagic flux with a GFP- or RFP-tagged LC3 processing assay.

To implement the tagged LC3 processing assay in mammalian cells, we searched for a protein tag that could be made resistant to lysosomal proteases and thus would not accumulate in lysosomes constitutively. Since ligand binding is known to increase a protein's thermal stability (*Celej et al., 2003*; *Vedadi et al., 2006*) and proteolytic resistance (*Stankunas et al., 2003*; *Park and Marqusee, 2005*; *Kaur et al., 2018*), we selected two protein tags that covalently bind with their respective ligands as candidate tags: Halo (*Los et al., 2008*) and SNAP-tag (SNAP) (*Keppler et al., 2003*).

HeLa cells stably expressing Halo-LC3B or SNAP-LC3B were exposed to their respective tetramethylrhodamine (TMR)-conjugated ligands for 20 min and then washed of the ligands. After 6 hr of starvation, a starvation-induced decrease in labeled and unlabeled Halo-LC3B was observed by immunoblotting and, for labeled Halo$^{TMR}$-LC3B, in-gel fluorescence detection (*Figure 1b*). Strikingly, free Halo was detected only for Halo$^{TMR}$-LC3B. The decrease in Halo-LC3B and Halo$^{TMR}$-LC3B levels and appearance of free Halo$^{TMR}$ were dependent on autophagy and lysosomal enzymatic activity as they were suppressed in *FIP200* KO cells (*Figure 1b–c*) and by the lysosomal inhibitor bafilomycin A$_1$ (*Klionsky et al., 2008*; *Mauvezin et al., 2015*; *Figure 1c*). Removal of the ligand was enough to prevent further labeling (*Figure 1—figure supplement 1*, compare lanes 8–10), avoiding the need for a blocking agent. In contrast, the generation of free SNAP was not detected from ligand-free and ligand-bound SNAP-LC3 during starvation (*Figure 1d*). Taken together, these data indicate that ligand-free Halo is susceptible to proteolysis but, unlike SNAP, Halo becomes resistant to proteolysis after covalently binding to its ligand and persists in lysosomes (*Figure 1e*). Therefore, autophagic flux can be quantitatively determined with pulse-chasable Halo-based reporters.

To assess the sensitivity and linearity of Halo-LC3B processing to the levels of autophagic activity, we obtained results for several time points between 0 and 8 hr of starvation. The amount of free Halo$^{TMR}$ generated from Halo$^{TMR}$-LC3B was detected as early as 0.5 hr of starvation and increased over time (*Figure 1f*). Calculation of the ratio of released Halo$^{TMR}$ to total Halo$^{TMR}$ (Halo$^{TMR}$-LC3B+Halo$^{TMR}$) showed a linear increase of free Halo$^{TMR}$ levels that was readily reproducible (*Figure 1g*). The advantages of the processing assay are: (1) pulse-labeling ensures that the starting amount of Halo$^{TMR}$-LC3B is relatively the same for each experimental condition (here, each time point), (2) the amount of free Halo is a positive readout (i.e. a signal is produced rather than reduced by autophagy), and (3) quantifying the amount of free Halo$^{TMR}$ produced is self-contained without the need for additional controls such as a lysosomal inhibitor control or a loading control. The Halo-LC3B processing assay is thus a simple and sensitive method with a wide linear range to quantitatively monitor autophagic flux in mammalian cells.

## HaloTag-based reporters to monitor autophagic flux by fluorescence microscopy

Next, we visualized the localization of mGFP-LC3B, mRFP-LC3B, and Halo-LC3B in HeLa cells. The number of mGFP-LC3B puncta increased during the first 2 hr of starvation but did not increase thereafter (*Figure 2a and c*), which is likely due to the quenching of mGFP signal in autolysosomes (*Kimura et al., 2007*). Under growing conditions, many mRFP-LC3B puncta were already present and their high rate of colocalization with LysoTracker revealed them to be lysosomes (*Figure 2a*), demonstrating mRFP's stability in acidic compartments (*Katayama et al., 2008*). Furthermore, an increase in the number of mRFP-LC3B puncta during starvation was not detected (*Figure 2a and c*), likely because the formation of mRFP-LC3B-marked autophagosomes was masked by the pre-existing intense lysosomal mRFP signal (*Katayama et al., 2008*). Contrastingly, Halo-LC3B that was labeled with ligand conjugated to fluorogenic SaraFluor 650T (SF650) demonstrated only a few faint punctate structures immediately after labeling, and starvation induced a continuous increase in the number of Halo$^{SF650}$-positive puncta over time (*Figure 2b and c*). Many of the Halo$^{SF650}$-positive puncta are

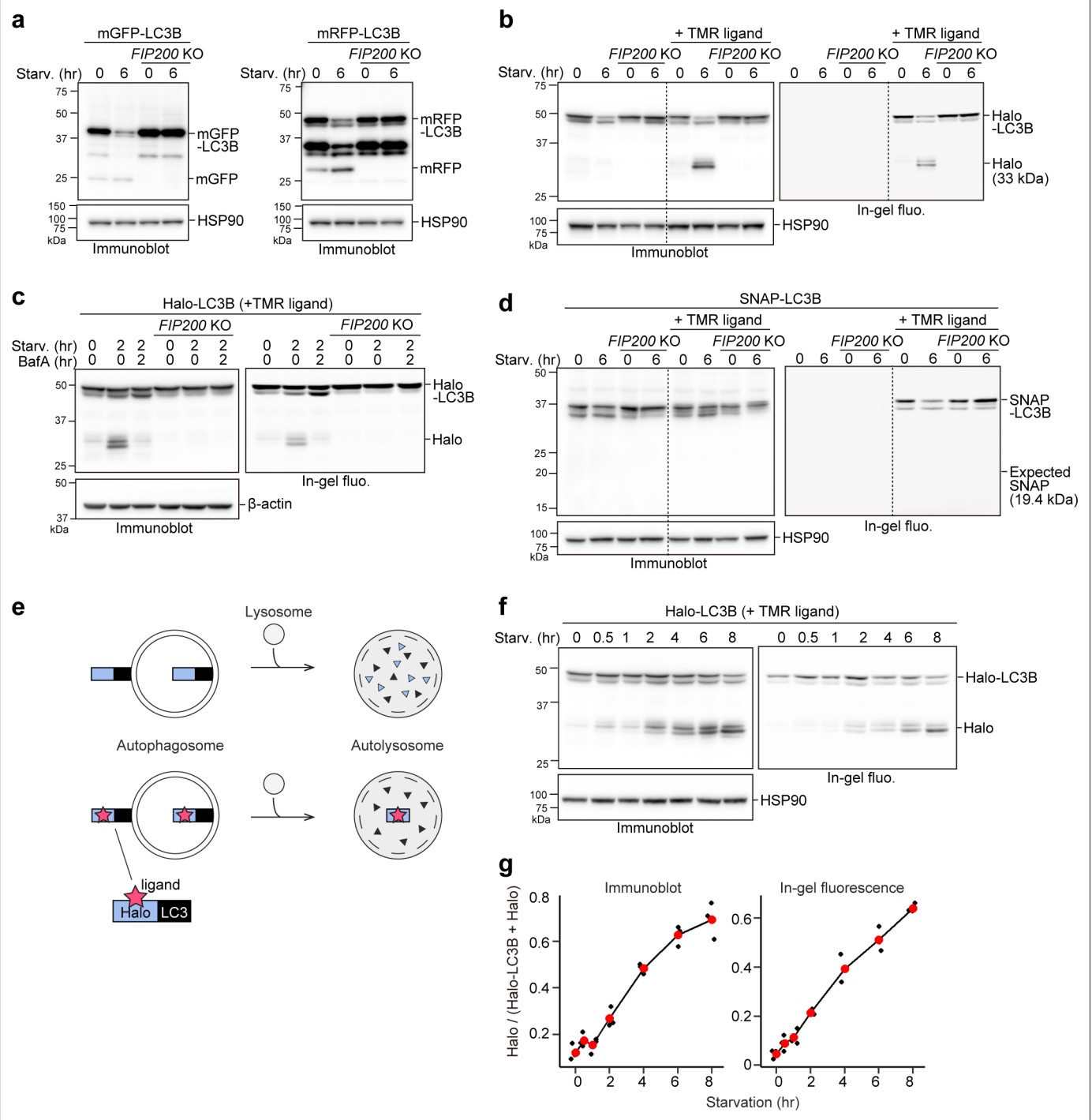

**Figure 1.** Processed ligand-bound HaloTag from HaloTag-LC3 is a quantifiable readout reflecting autophagic flux. (**a**) Immunoblotting of total cell lysates from wild-type and *FIP200* knockout (KO) HeLa cells stably expressing monomeric GFP (mGFP)-LC3B or monomeric RFP (mRFP)-LC3B that was in nutrient-rich medium (0 hr) or incubated for 6 hr in starvation medium. (**b**) Immunoblotting and in-gel fluorescence detection of total cell lysates from wild-type and *FIP200 KO* HeLa cells stably expressing Halo-LC3B that was pulse-labeled for 20 min with 100 nM of tetramethylrhodamine (TMR)-conjugated ligand in nutrient-rich medium. The cells were immediately collected (0 hr) or incubated in starvation medium for 6 hr. (**c**) Same cells and labeling procedure as in (**b**) except that the cells were collected or incubated in starvation medium with or without 100 nM bafilomycin A₁ (BafA) for 2 hr. (**d**) Same labeling procedure as in (**b**) except with HeLa cells stably expressing SNAP-tag (SNAP)-LC3B and TMR-conjugated ligand targeting SNAP. (**e**) Illustration of the fate of ligand-free and ligand-bound Halo-LC3B in autolysosomes. Halo-LC3B is processed by lysosomal hydrolases, releasing Halo from LC3B. When free from ligand, Halo is unstable and quickly degraded in lysosomes like LC3B. In contrast, ligand-bound Halo is stable against degradation and accumulates in lysosomes. Released Halo^ligand is detectable by immunoblot and, if a fluorescent ligand was used, in-gel fluorescence

*Figure 1 continued on next page*

*Figure 1 continued*

detection. The amount of free Halo$^{ligand}$ separated from LC3B reflects the level of autophagic flux. (**f**) Immunoblotting and in-gel fluorescence detection of total cell lysates from wild-type HeLa cells stably expressing Halo-LC3B, pulse-labeled with 100 nM TMR-conjugated ligand in nutrient-rich medium for 20 min, and starved for the indicated durations. (**g**) Quantification of results shown in (**f**). Halo$^{TMR}$ band intensity was normalized by the sum of the band intensities Halo$^{TMR}$-LC3B and Halo$^{TMR}$. Mean values of data from three experiments are shown with red points that are traced by a line.

The online version of this article includes the following source data and figure supplement(s) for figure 1:

**Source data 1.** Uncropped blot images of *Figure 1a, b, c, d and f*, and *Figure 1—figure supplement 1*.

**Figure supplement 1.** HaloTag-based processing assay can be conducted without a blocking agent.

autolysosomes as they colocalized with LysoTracker (*Figure 2b*) and the lysosomal membrane protein LAMP1 (*Figure 2—figure supplement 1a*), which shows that Halo$^{SF650}$ is stable in acidic lysosomes. Although autophagosomes cannot be distinguished from autolysosomes with Halo-LC3, the number of ligand-bound Halo puncta formed following pulse-labeling of Halo-based reporters can indicate autophagic flux.

The minimal lysosomal background signal of the Halo-based assay prompted us to use Halo instead of mRFP in the established mRFP-mGFP-LC3B reporter (*Figure 2d*; *Kimura et al., 2007*). This reporter was originally developed to distinguish between mRFP-positive mGFP-positive (mRFP$^+$mGFP$^+$) auto-phagosomes and mRFP-positive mGFP-negative (mRFP$^+$mGFP$^-$) autolysosomes, but the constitutive accumulation of mRFP in lysosomes often hinders the detection of newly generated autolysosomes (mRFP$^+$mGFP$^-$) (*Figure 2—figure supplement 1b and c*). In contrast, Halo$^{SF650}$-mGFP-LC3 was mostly cytosolic immediately after labeling (*Figure 2e*, *Figure 2—figure supplement 1d*). Starvation induced the formation of Halo$^{SF650+}$mGFP$^+$ puncta that represent autophagosomes (arrowheads) and Halo$^{SF650+}$mGFP$^-$ puncta that represent autolysosomes (arrows) as confirmed by their colocalization with LysoTracker or LAMP1-mRuby3 (*Figure 2e and f*, *Figure 2—figure supplement 1d*). Adding the lysosomal inhibitor bafilomycin A$_1$ resulted in the accumulation of Halo$^{SF650+}$ and mGFP$^+$ in autopha-gosomes and autolysosomes (*Figure 2e and f*, *Figure 2—figure supplement 1d*). Hence, the lack of pre-existing lysosomal accumulation makes Halo-mGFP-LC3B a preferable alternative to mRFP-mGFP-LC3B as a fluorescence microscopy-based autophagic flux reporter.

We also generated Halo-mGFP-LC3B-mRFP as a modified version of the EGFP-LC3B-mRFP reporter (*Kaizuka et al., 2016*). With the original EGFP-LC3B-mRFP reporter, autophagic flux can be estimated from GFP:RFP fluorescence ratio because this reporter is cleaved into EGFP-LC3B and mRFP by ATG4 and only EGFP-LC3B is degraded by autophagy since mRFP remains in the cytosol (*Kaizuka et al., 2016*). Adding Halo at the N-terminus of mGFP-LC3B-mRFP yielded a reporter that can be used in the processing assay (*Figure 2—figure supplement 1e*), the autophagosome/autolysosome formation assay (*Figure 2—figure supplement 1f*), and the GFP:RFP ratiometry assay (*Figure 2—figure supple-ment 1g*). Thus, Halo-mGFP-LC3B-mRFP can be used as a stably expressed 'three-in-one' reporter.

In summary, pulse-chasable Halo-based reporters are suitable for monitoring autophagic flux by fluorescence microscopy in addition to the processing assay readout.

## The HaloTag-based processing assay can be adapted to monitor selective autophagy pathways

A major advantage of the yeast GFP-Atg8 processing assay is its versatility: Atg8 can be replaced with a protein of interest to monitor the activity of selective autophagy, such as endoplasmic retic-ulum (ER)-phagy and mitophagy. Hence, we prepared Halo-mGFP-KDEL that localizes to the ER lumen (*Figure 3—figure supplement 1a*) and pSu9-Halo-mGFP that localizes to the mitochondrial matrix (*Figure 3—figure supplement 1b*). Generation of the 33 kDa Halo$^{TMR}$ band from pulse-labeled Halo$^{TMR}$-mGFP-KDEL was detected in a FIP200-dependent manner upon starvation (*Figure 3a and b*). Production of free Halo$^{TMR}$ was a clear indicator of ER-phagy than the reduction in ER protein levels (*Figure 3a–b* vs. *Figure 3c-d*). With fluorescence microscopy, ER-phagy flux was detected by the formation of Halo$^{SF650+}$mGFP$^-$ puncta that colocalized with LysoTracker (*Figure 3e and f*) and LAMP1 (*Figure 3—figure supplement 1c*). Similarly, Parkin-mediated mitophagic activity induced by oligomycin and antimycin treatment was detected by pSu9-Halo-mGFP processing (*Figure 3g and h*) as well as the formation of Halo$^{SF650+}$mGFP$^-$ puncta colocalized with the lysosome (*Figure 3i and j*, *Figure 3—figure supplement 1d*). An additional 37 kDa HaloTag band was observed in the SDS-PAGE

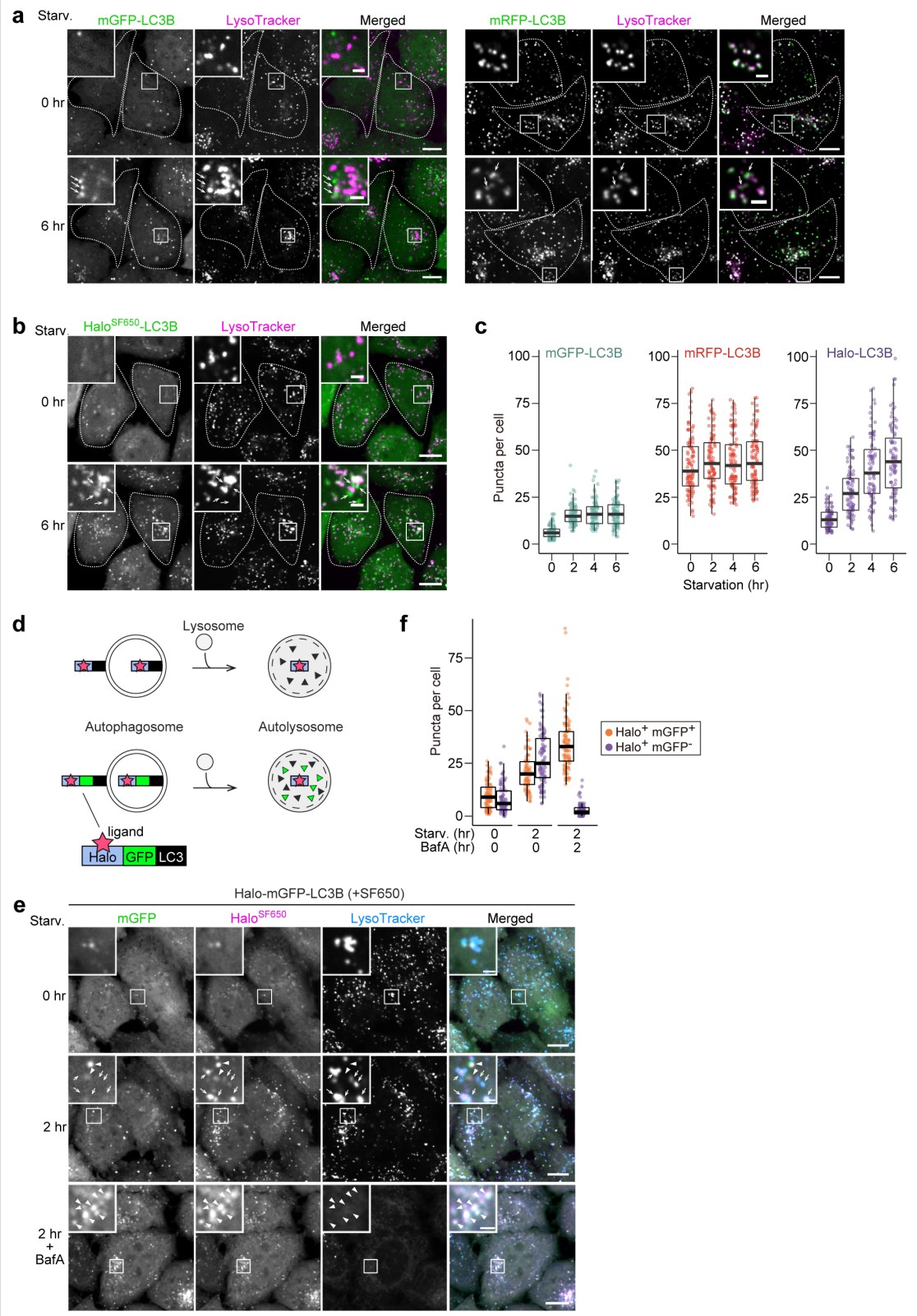

**Figure 2.** HaloTag-based reporters can be used to examine autophagic flux by fluorescence imaging. (**a**) Fluorescence images of wild-type HeLa cells stably expressing mGFP-LC3B or mRFP-LC3B in nutrient-rich medium or after 2 hr in starvation medium containing 75 nM LysoTracker Deep Red. (**b**) Fluorescence images of wild-type HeLa cells stably expressing HaloTag (Halo)-LC3B labeled with SF650-conjugated ligand. The cells were in nutrient-rich medium or incubated for 2 hr in starvation medium containing 200 nM of SF650-conjugated ligand and 75 nM LysoTracker Red. Arrows

*Figure 2 continued on next page*

*Figure 2 continued*

point to Halo puncta without LysoTracker signal that represent autophagosomes. (**c**) Quantification of mGFP, mRFP, or Halo$^{SF650}$ puncta in the cells of (**a**) and (**b**). n=87–99 cells. (**d**) Schematic of fluorescence changes of fluorescent ligand-bound Halo-LC3B and Halo-mGFP-LC3B in autophagosomes and autolysosomes. Halo$^{ligand}$ stays fluorescent and stable, whereas mGFP is quenched and degraded in autolysosomes. With Halo-mGFP-LC3B, autophagosomes appear as double-positive (Halo$^{ligand}$ and mGFP) puncta while autolysosomes appear as single-positive (Halo$^{ligand}$ only) puncta. (**e**) Fluorescence images of wild-type HeLa cells stably expressing Halo-mGFP-LC3B labeled with SF650-conjugated ligand. The cells were growing, starved, or starved with 100 nM bafilomycin A$_1$ (BafA) for 2 hr in medium containing 200 nM of SF650-conjugated ligand and 75 nM LysoTracker Red. Arrowheads point to Halo$^+$mGFP$^+$ puncta that represent autophagosomes; arrows point to Halo$^+$mGFP$^-$ puncta that represent autolysosomes. (**f**) Quantification of Halo$^{SF650+}$mGFP$^+$ puncta representing autophagosomes and Halo$^{SF650+}$mGFP$^-$ puncta representing autolysosomes in cells shown in (**e**). n=86–101 cells. Scale bar = 10 µm (main), 2 µm (magnified images) (**a, b, e**). In box plots, solid bars indicate medians, boxes indicate the interquartile range (25th–75th percentile), and whiskers indicate the largest and smallest values within 1.5 times the interquartile range (**c, f**).

The online version of this article includes the following source data and figure supplement(s) for figure 2:

**Figure supplement 1.** HaloTag can be combined with GFP- and/or RFP-based autophagy reporters.

**Figure supplement 1—source data 1.** Uncropped blot images of *Figure 2—figure supplement 1e*.

based results (*Figure 3a and g*) that is most likely a cleavage product of the HaloTag constructs. The lack of cytosolic Halo and GFP signal (*Figure 3e and i*) indicates that this still-fluorescent product (in-gel fluorescence images in *Figure 3a and g*) localizes to the targeted organelle and would not result in background signal that might interfere with result interpretation. Therefore, the Halo-based assays can be adapted to monitor selective autophagic flux.

## Bulk nonselective autophagic flux can be detected with the HaloTag-based processing assay

Finally, we examined whether the Halo-based processing assay could be used to detect bulk nonselective autophagic flux, which more accurately reflects total autophagic flux compared to LC3-based methods since LC3 and GABARAP family proteins are preferentially degraded by autophagy (*Klionsky et al., 2021b*). To monitor nonselective clearance of cytosolic material, we constructed Halo-mGFP, which should be taken up randomly into autophagosomes or lysosomes like the yeast Pgk1-GFP reporter (*Welter et al., 2010*).

Despite occurring at much lower levels than LC3-dependent autophagic flux detected by Halo$^{TMR}$-LC3B, the lack of lysosomal background allowed us to detect the generation of free Halo$^{TMR}$ from pulse-labeled Halo$^{TMR}$-mGFP (*Figure 4a*). Accordingly, we were able to track the progression of starvation-induced bulk nonselective autophagic flux over time by Halo$^{TMR}$-mGFP processing (*Figure 4b and c*) and by visualizing the gradual accumulation of Halo$^{SF650}$ signal in the lysosomes of cells stably expressing Halo-mGFP labeled with SF650 ligand (*Figure 4d and e*, *Figure 4—figure supplement 1a and b*). Halo$^{TMR}$-mGFP processing and Halo$^{SF650}$ puncta formation were not observed in *FIP200* KO cells (*Figure 4a and d*). In contrast, Halo$^{TMR}$-mGFP processing was slightly detected in *ATG3* KO and *ATG5* KO cells (*Figure 4a*), which is consistent with our previous finding that the ATG conjugation systems are not completely essential in mammalian cells (*Tsuboyama et al., 2016*). These results show that the Halo-GFP processing assay can be used to monitor bulk nonselective autophagic flux, even in cells lacking ATG conjugation system components.

## Discussion

It is generally difficult to implement the autophagic flux assays currently available for mammalian cells. In this study, we showed that the Halo-based processing assay is a simple yet effective way to quantitatively monitor various forms of autophagic flux in mammalian cells, from LC3-dependent autophagic flux to bulk nonselective autophagy. While this manuscript was being prepared, an independent study on the use of Halo-based reporters to monitor autophagic flux was published (*Rudinskiy et al., 2022*). In contrast to our main aim of developing a reporter processing assay for mammalian cells, Rudinskiy et al. sought to overcome the issue of lysosomal RFP accumulation with Halo-based reporters that can be fluorescently 'activated' upon binding to fluorescent ligands. Our study demonstrated that the 'activation' of Halo-based reporters is achieved not only by the introduction of fluorescent Halo ligand but, importantly, also by the stabilization of Halo against proteolysis after ligand binding.

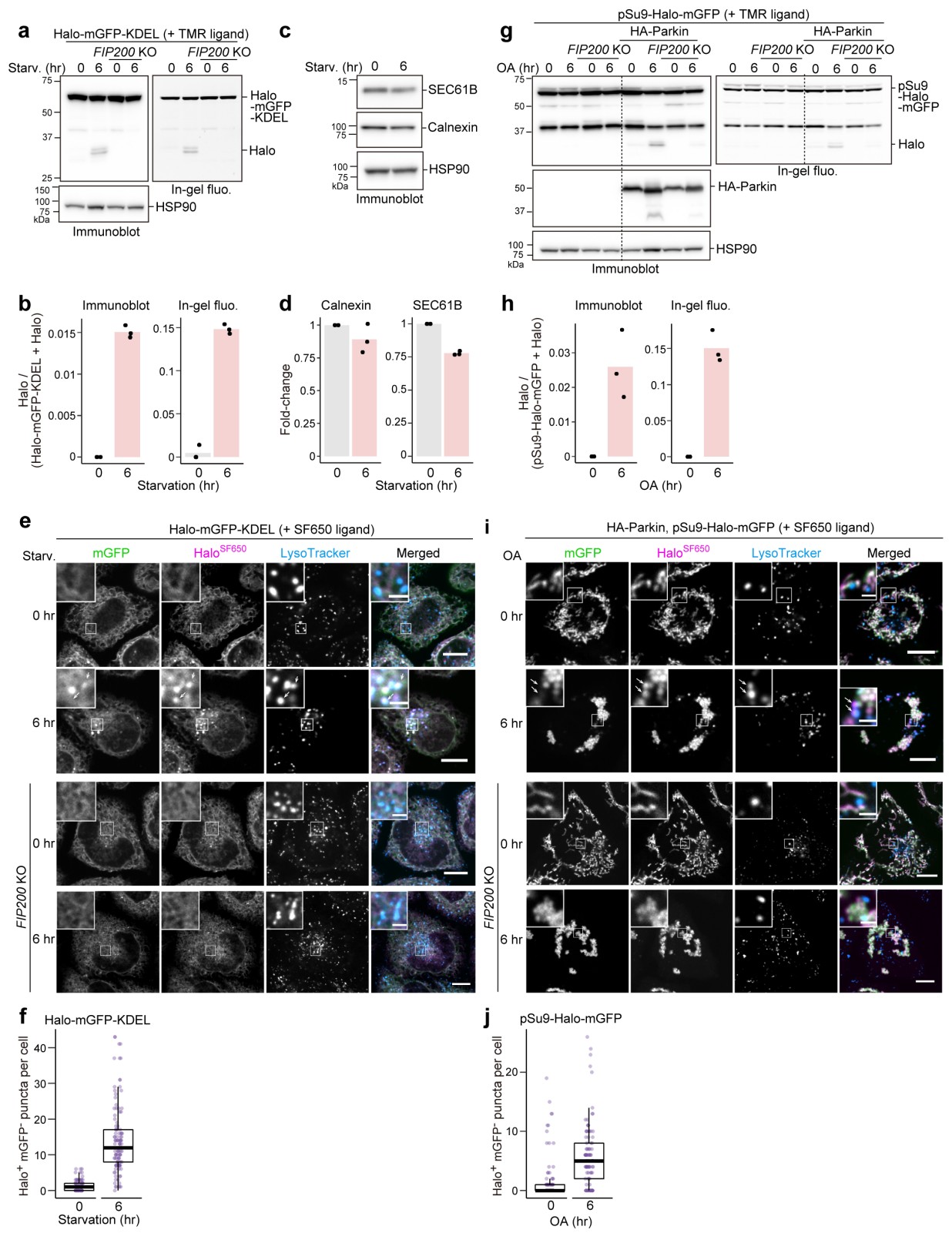

**Figure 3.** HaloTag-based reporters can be adapted to monitor selective autophagy. (**a**) Immunoblotting and in-gel fluorescence detection of wild-type and *FIP200* knockout (KO) HeLa cells stably expressing HaloTag (Halo)-mGFP-KDEL pulse-labeled for 20 min with 100 nM tetramethylrhodamine (TMR)-conjugated ligand in nutrient-rich medium. The cells were collected or incubated for 6 hr in starvation medium. (**b**) Quantification of Halo band intensities in (**a**). n=3. (**c**) Immunoblotting of total cell lysates from wild-type HeLa cells in nutrient-rich medium or after 6 hr in starvation medium.

*Figure 3 continued on next page*

*Figure 3 continued*

(**d**) Quantification of band intensities for the indicated proteins in (**c**). n=3. (**e**) Fluorescence images of the cells described in (**a**) under nutrient-rich conditions or for 6 hr under starvation conditions. The media contained 200 nM SF650-conjugated ligand and 75 nM LysoTracker Red. Arrows point to Halo$^{SF650+}$mGFP$^-$ puncta representing autolysosomes. (**f**) Quantification of Halo$^{SF650+}$mGFP$^-$ puncta that represent autolysosomes in cells shown in (**e**). n=111–115 cells. (**g**) Immunoblotting and in-gel fluorescence detection of total cell lysates from wild-type or *FIP200* KO HeLa cells stably expressing pSu9-Halo-mGFP only or together with HA-Parkin. After pulse-labeling with 100 nM TMR ligand in nutrient-rich medium, the cells were collected or incubated for 6 hr in nutrient-rich medium containing 1 μM oligomycin and 2 μM antimycin (OA collectively). (**h**) Quantification of Halo band intensities in (**g**). (**i**) Fluorescence images of wild-type and *FIP200* KO HeLa cells stably expressing pSu9-Halo-mGFP and HA-Parkin. The cells were in nutrient-rich medium containing 200 nM SF650 ligand, 75 nM LysoTracker Red, with or without OA. Arrows point to Halo$^{SF650+}$mGFP$^-$ puncta that represent autolysosomes. (**j**) Quantification of Halo$^{SF650+}$mGFP$^-$ puncta in cells shown in (**i**). n=99 cells. Scale bar = 10 μm (main), 2 μm (magnified images) (**e, i**). In box plots, solid bars indicate medians, boxes indicate the interquartile range (25th–75th percentile), and whiskers indicate the largest and smallest values within 1.5 times the interquartile range (**f, j**).

The online version of this article includes the following source data and figure supplement(s) for figure 3:

**Source data 1.** Uncropped blot images of *Figure 3a, c and g*.

**Figure supplement 1.** Halo-GFP-KDEL and pSu9-Halo-GFP localize to endoplasmic reticulum (ER) and mitochondria, respectively.

The most significant advantage the HaloTag reporters have over currently available ATG8-based reporters and autophagy assays is that they can be used in a pulse-chase manner. This allows for minimal background noise since unlabeled reporters will not accumulate within lysosomes, and for the readout of free Halo to be unambiguously autophagy-dependent since it occurs only when the reporter has been labeled, delivered into, and processed in lysosomes. Another major advantage is the dispensation of a lysosome inhibitor control that may affect transcription/translation. Furthermore, reporter processing can be detected by immunoblotting, which is a widely available and commonly used laboratory technique. It can also be detected by in-gel imaging if available, which would reduce the time required to obtain results. Last, by replacing the partner protein of Halo, users can easily adapt the assay to monitor their autophagy pathway of interest or the autophagic clearance of a protein of interest.

The assay does have limitations. One that is inevitable is the need for exogenous expression. For experiments that are sensitive to variation in expression levels, clonal isolation may be required. Furthermore, the cost of Halo ligands might be prohibitive. A more cost-effective alternative could be the non-fluorescent Halo ligands, such as 7-bromoheptanol, which were developed as blocking agents (*Merrill et al., 2019*), and Halo band detection would be achieved only by immunoblotting. Another limitation is the difficulty in quantifying the absolute rate of reporter incorporation into lysosomes precisely due to the gradual degradation of even Halo$^{ligand}$. Additionally, the need to electrophoretically separate the full-length reporter from the free Halo$^{ligand}$ band limits the use of this assay for large-scale screening. However, the good signal-to-noise ratio of Halo reporters in fluorescent imaging makes them ideal reporters for microscopy-based high-content screening in the same manner as was achieved with GFP-LC3 (*Orvedahl et al., 2011*; *McKnight et al., 2012*; *Koepke et al., 2020*) and dual-fluorescent Rosella-LC3 (*Arias-Fuenzalida et al., 2019*). Finally, Halo-LC3 reporter processing is not suitable for monitoring the activity of non-canonical forms of autophagy, where LC3 is conjugated to single-membrane organelles (e.g. LC3-associated phagocytosis; *Heckmann and Green, 2019*; *Herb et al., 2020*) as the reporter will not be delivered into lysosomes.

We anticipate the Halo-based processing assay to be useful to most studies on mammalian autophagy and lysosome-dependent protein clearance. With Halo-GFP, researchers can investigate the contribution of autophagy proteins by examining whether altering a protein by depletion/inhibition/mutation will affect nonselective autophagic flux. This may be especially useful to researchers investigating proteins that are directly required for commonly used autophagic flux assays, such as factors involved in the LC3 or GABARAP conjugation systems. Indeed, we managed to detect low levels of autophagic flux in *ATG3* KO and *ATG5* KO cells (*Figure 4a*), which cannot be detected by conventional LC3-based assays. This result is consistent with the recent consensus that, although severely reduced, autophagic flux is not completely blocked in cells lacking the ATG conjugation systems (reviewed in *Mizushima, 2020*). Halo-GFP could also be potentially used to study microautophagy. Furthermore, Halo-based reporters have been successfully employed in animal models (*England et al., 2015*) such as mice (*Masch et al., 2018*; *Rivas-Pardo et al., 2020*) and zebrafish (*Wan et al., 2019*), which

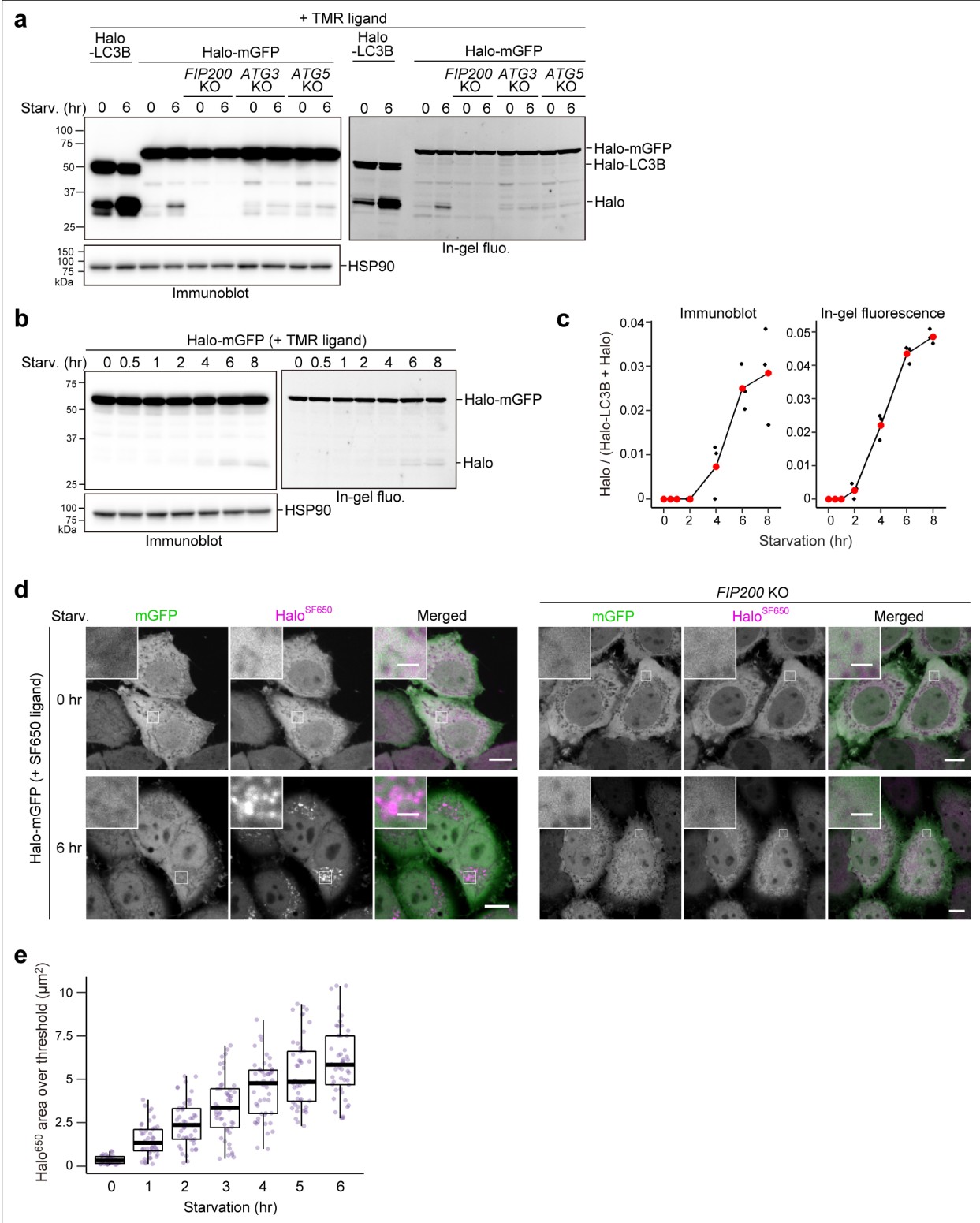

**Figure 4.** Bulk nonselective autophagic flux can be detected with HaloTag-GFP. (**a,b**) Immunoblotting and in-gel fluorescence detection of total cell lysates from wild-type, *FIP200* knockout (KO), *ATG3* KO, and *ATG5* KO HeLa cells stably expressing HaloTag (Halo)-LC3B (**a**) or Halo-mGFP (**a,b**). After pulse-labeling for 20 min with 100 nM tetramethylrhodamine (TMR)-conjugated ligand in nutrient-rich medium, the cells were collected or incubated for 6 hr (**a**) or the indicated times (**b**) in starvation medium. (**c**) Quantification of time course results shown in (**b**). Mean values of three experiments are shown with red points that are traced with a line. (**d**) Fluorescence images of wild-type and *FIP200* KO HeLa cells stably expressing Halo-mGFP in nutrient-rich medium or after 6 hr of starvation. The media contained 200 nM of SF650-conjugated ligand. Scale bar = 10 µm (main), 2 µm (magnified

*Figure 4 continued on next page*

*Figure 4 continued*

images). (**e**) Quantification of the area of Halo$^{SF650}$ signal that exceeded a 98th percentile intensity threshold in cells described in (**d**) at the indicated durations of starvation. n=42–48 cells. In box plots, solid bars indicate medians, boxes indicate the interquartile range (25th–75th percentile), and whiskers indicate the largest and smallest values within 1.5 times the interquartile range.

The online version of this article includes the following source data and figure supplement(s) for figure 4:

**Source data 1.** Uncropped blot images of *Figure 4a and b*.

**Figure supplement 1.** Bulk nonselective autophagic flux can be detected with Halo-GFP.

suggest that the processing assay could be an additional method for assessing in vivo autophagic flux, e.g. when comparing autophagic flux among tissues.

In conclusion, the Halo-based processing assay can be used to monitor mammalian autophagic flux quantitatively and reliably with relative ease. The Halo-based reporters can also be readily observed by fluorescence microscopy. Besides serving as an alternative to conventional autophagic flux assays, the Halo-based assay has a myriad of potential applications that may lead to the discovery of new aspects and thus a deeper understanding of mammalian autophagy.

## Materials and methods

### Cell lines and culture conditions

HeLa cells, human embryonic kidney (HEK) 293T cells, and mouse embryonic fibroblasts (MEFs) authenticated by RIKEN were used in this study. Cells were maintained in Dulbecco's Modified Eagle Medium (DMEM; Sigma-Aldrich, D6546) supplemented with 10% fetal bovine serum (FBS; Sigma-Aldrich, 173012) and 2 mM L-glutamine (Gibco, 25030–081) in a 5% $CO_2$ incubator at 37°C. *FIP200* KO HeLa cells (*Tsuboyama et al., 2016*) and *Fip200* KO MEFs *Gan et al., 2006* have been described previously.

### Plasmids

Plasmids for stable expression in HeLa cells were generated as follows: DNA fragments encoding monomeric enhanced GFP with A206K mutation (mGFP), monomeric RFP (mRFP), mRuby3 (codon-optimized from Addgene #74252), HaloTag7 (Promega, N2701), or SNAP-tag (New England BioLabs, N9181S) were inserted into the retroviral plasmids pMRX-IP (harboring a puromycin-resistant marker; *Kitamura et al., 2003*; *Saitoh et al., 2003*), pMRX-IB (harboring a blasticidin-resistant marker; *Morita et al., 2018*), or pMRX-No (without a drug resistance marker) by the seamless ligation cloning extract (SLiCE) method (*Motohashi, 2017*). Then, DNA fragments encoding rat LC3B, the signal sequence of *Drosophila* BiP (residues 1–18) and KDEL (for pMRX-IB-Halo-mGFP-KDEL), or the presequence of *Neurospora crassa* Fo-ATPase subunit 9 (residues 1–69; for pMRX-IB-pSu9-Halo-mGFP) were inserted into the pMRX-IP-, pMRX-IB-, pMRX-No-based plasmids by the SLiCE method. mRFP-GFP-LC3B in pMXs was described previously (*Jiang et al., 2014*). Primers used in this study are listed in *Supplementary file 1*. Plasmids containing the Halo constructs used in this study can be found in addgene: 184899 (Halo-LC3), 184901 (Halo-mGFP-rLC3), 184902 (Halo-mGFP), 184904 (Halo-mGFP-KDEL), and 184905 (pSu9-Halo-mGFP).

### Stable expression in HeLa cells and MEFs by retrovirus infection

To prepare the retrovirus solution, HEK293T cells were transfected for 4–6 hr with the pMRX-IP-based, pMRX-IB-based, or pMXs-based retroviral plasmid (*Kitamura et al., 2003*; *Saitoh et al., 2003*), pCG-gag-pol, and pCG-VSV-G using Lipofectamine 2000 (Thermo Fisher Scientific, 11668019), following which the medium was replaced with DMEM. After 2–3 days, the retrovirus-containing medium was harvested, filtered with a 0.45 µm filter unit (Millipore, Ultrafree-MC), and added to HeLa cells with 8 µg/mL polybrene (Sigma-Aldrich, H9268). After a day, selection was performed with 1–2 µg/mL puromycin (Sigma-Aldrich, P8833) or 2–3 µg/mL blasticidin (Fujifilm Wako Pure Chemical Corporation, 022–18713).

## Antibodies and reagents

Primary antibodies used in this study are as listed: mouse monoclonal anti-Halo (Promega, G9211), rabbit polyclonal anti-SNAP-tag (New England BioLabs, P9310S), mouse monoclonal anti-HSP90 (BD Transduction Laboratories, 610419), anti-β-actin (Sigma-Aldrich, A2228), rabbit polyclonal anti-LC3 (*Kabeya et al., 2000*), rabbit polyclonal anti-GFP (Thermo Fisher Scientific, A6455), and mouse monoclonal anti-RFP (MBL, M155-3). Secondary antibodies are HRP-conjugated goat polyclonal anti-rabbit IgG (Jackson ImmunoResearch Laboratories, 111-035-144) and HRP-conjugated goat polyclonal anti-mouse IgG (Jackson ImmunoResearch Laboratories, 115-035-003). LysoTracker Red DND-99 or Deep Red (Thermo Fisher Scientific, L7528, L12492) were applied at 75 nM.

## Protein extraction

Cells were incubated in DMEM with 100 nM TMR-conjugated Halo ligand (Promega, G8251) for 20 min. The cells were then collected for protein extraction or washed twice with phosphate-buffered saline (PBS) and incubated with amino acid-free and FBS-free DMEM (Wako Pure Chemical Industries, 048–33575) to induce autophagy by starvation or to DMEM with 1 μM oligomycin (Merck Millipore, 495455) and 2 μM antimycin (Sigma-Aldrich, A8674) to induce mitophagy. After the desired incubation time, cells were scraped into ice-cold PBS and centrifuged at 2000×g for 2 min. The cell pellet was then resuspended in lysis buffer (25 mM HEPES-KOH [pH 7.2], 150 mM NaCl, 2 mM MgSO$_4$, 0.2% $n$-dodecyl-β-D-maltoside [Nacalai Tesque, 14239–54] with protease inhibitor [Sigma-Aldrich, P8340]). After 20 min of incubation on ice, 1/10 volume of lysis buffer containing benzonase (Merck Millipore, 70664) was added to the cell suspension to a final 1/200 dilution of benzonase. Protein concentration was determined with NanoDrop One spectrophotometer (Thermo Fisher Scientific).

## In-gel fluorescence imaging and immunoblotting

For each sample, 20 μg of protein was separated by SDS-PAGE. For in-gel fluorescence imaging, the gel was immediately visualized with FUSION SOLO.7S.EDGE (Vilber-Lourmat) after SDS-PAGE. For immunoblotting, the samples were transferred from the SDS-PAGE gel to Immobilon-P polyvinylidene difluoride membranes (Millipore, WBKLS0500) with Trans-Blot Turbo Transfer System (Bio-Rad). After incubation with the relevant antibody, the signals from incubation with SuperSignal West Pico Chemiluminescent Substrate (Thermo Fisher Scientific, 34579) or Immobilon Western Chemiluminescent HRP Substrate (Millipore, WBKLS0500) were detected with FUSION SOLO.7S.EDGE (Vilber-Lourmat). Band intensities were measured with *Gel Analyzer* in the open-source image processing software Fiji (*Schindelin et al., 2012*).

## Fluorescence imaging (live-cell)

Imaging was conducted with the Olympus SpinSR10 spinning-disk confocal super-resolution microscope equipped with a Hamamatsu ORCA-Flash 4.0 camera, a UPLAPO OHR 60× (NA 1.50) lens, and the SORA disk in place. The microscope was operated with the Olympus cellSens Dimension 2.3 software.

HeLa cells or MEFs were seeded onto a four-chamber glass-bottom dish (Greiner Bio-One) at least 48 hr before imaging. 20 min before image acquisition, the cells were incubated with 200 nM SF650-conjugated Halo ligand (GoryoChemical, A308-02) in FluoroBrite DMEM (Thermo Fisher Scientific, A1896701) supplemented with 10% FBS, 2 mM L-glutamine, and 50 U/ml penicillin and 50 μg/ml streptomycin (Gibco, 15070–063). After acquiring images of the cells under growing conditions, they were washed with PBS twice and then incubated in amino acid-free and FBS-free DMEM (Wako Pure Chemical Industries, 048–33575) with 200 nM SF650-conjugated Halo ligand to induce autophagy by starvation or in the aforementioned supplemented FluoroBrite DMEM with 200 nM SF650-conjugated Halo ligand, 1 μM oligomycin, and 2 μM antimycin to induce mitophagy.

Images were processed using the open-source image processing software Fiji (*Schindelin et al., 2012*). Cells were processed individually: each cell was first isolated as a single image, which was later processed. In *Figure 2a and b*, puncta were identified and counted with *White Top Hat* (from MorphoLibJ *Legland et al., 2016*) followed by *Analyze Particles*. For double-positive puncta (RFP$^+$GFP$^+$ or Halo$^+$GFP$^+$) and single-positive (RFP$^+$GFP$^-$ or Halo$^+$GFP$^-$) puncta identification in *Figures 2e, 3e and i*, and *Figure 2—figure supplement 1b*, the positions of the puncta were identified with *Find Maxima* and marked as points on a binary image for both fluorescence channels. With the binarized

images, double-positive puncta were identified with the *AND* function in *Image Calculator* (i.e. 'GFP' image *AND* 'RFP' image) while single-positive puncta were identified with the *Subtract* function in *Image Calculator* (i.e. 'RFP' image *Subtract* 'GFP' image). The resulting puncta after each operation were counted with *Analyze Particles*. In *Figure 4e and a* 98th percentile threshold was set and the area of the signal remaining after thresholding was measured with *Analyze Particles*.

## Flow cytometry

HeLa cells or MEFs were treated with or without 250 nM Torin1 (Tocris Bioscience, 4247) in DMEM for 24 hr. Each sample was detached by trypsinization, added to an equal volume of DMEM to inactivate trypsin, and centrifuged at 2000×g for 2 min at 4°C. The cell pellet was then resuspended in ice-cold PBS containing 7-amino-actinomycin D (BD Pharmingen, 559925) diluted 1:100. After 10 min on ice, five volumes of ice-cold PBS were added (e.g. 500 μl to 100 μl) and the sample was centrifuged at 2000×g for 2 min at 4°C. The cell pellet was resuspended in ice-cold PBS and subjected to flow cytometry by a cell analyzer (Sony, EC800) equipped with 488 nm and 561 nm lasers. All data points were transferred to Kaluza Analysis 2.1 software (Beckman Coulter) for analysis.

## Data and material availability

All data generated or analyzed during this study are included in the manuscript and supporting files. Plasmids and plasmid maps for the constructs stated in this manuscript will be available from *Addgene* (https://www.addgene.org/Noboru_Mizushima/).

## Acknowledgements

We are grateful to Maya Shirakawa for technical assistance with the experiments; Shoji Yamaoka for the pMRX-IP plasmid; and Teruhito Yasui for the pCG-gag-pol and pCG-VSV-G plasmids. This work was supported by the Exploratory Research for Advanced Technology (ERATO) research funding program of the Japan Science and Technology Agency (JST) (JPMJER1702 to NM) and a Grant-in-Aid for Transformative Research Areas (A) (21H05256 to HY) and for Specially Promoted Research (22H04919 to NM) from the Japan Society for the Promotion of Science (JSPS). WWY was supported by the Japanese Ministry of Education, Culture, Sports, Science and Technology.

## Additional information

### Competing interests

Noboru Mizushima: Reviewing editor, *eLife*. The other authors declare that no competing interests exist.

### Funding

| Funder | Grant reference number | Author |
| --- | --- | --- |
| Japan Science and Technology Agency | Exploratory Research for Advanced Technology (ERATO)/ JPMJER1702 | Noboru Mizushima |
| Japan Society for the Promotion of Science | Transformative Research Areas (A)/ 21H05256 | Hayashi Yamamoto |
| Japan Society for the Promotion of Science | Specially Promoted Research/ 22H04919 | Noboru Mizushima |

The funders had no role in study design, data collection and interpretation, or the decision to submit the work for publication.

### Author contributions

Willa Wen-You Yim, Data curation, Formal analysis, Investigation, Writing – original draft, Writing – review and editing; Hayashi Yamamoto, Conceptualization, Data curation, Formal analysis, Supervision, Funding acquisition, Investigation, Methodology, Writing – original draft, Project administration,

Writing – review and editing; Noboru Mizushima, Supervision, Funding acquisition, Writing – original draft, Project administration, Writing – review and editing

### Author ORCIDs
Hayashi Yamamoto http://orcid.org/0000-0002-2831-1463
Noboru Mizushima http://orcid.org/0000-0002-6258-6444

### Decision letter and Author response
Decision letter https://doi.org/10.7554/eLife.78923.sa1
Author response https://doi.org/10.7554/eLife.78923.sa2

## Additional files

### Supplementary files
- MDAR checklist
- Supplementary file 1. Sequences of the primers used for plasmid construction.

### Data availability
All data generated or analysed during this study are included in the manuscript and supporting file; Source Data files have been provided for Figures 1, S1, S2, 3, and 4.

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
