## [Editor Report]

This paper describes the development of quantitative approaches to measure the turnover of cellular material in the lysosome. These new tools are carefully validated and the data shows their application to several physiologically relevant types of autophagy. This assay will be very useful for any researcher interested in studying autophagy flux

---

## [Decision Letter]

**Decision letter after peer review:**

Thank you for submitting your article "A pulse-chasable reporter processing assay for mammalian autophagic flux with HaloTag" for consideration by *eLife*. Your article has been reviewed by 3 peer reviewers, one of whom is a member of our Board of Reviewing Editors, and the evaluation has been overseen by Vivek Malhotra as the Senior Editor. The following individual involved in the review of your submission has agreed to reveal their identity: Sharon Tooze (Reviewer #3).

Essential revisions:

1) Modify the text to address reviewer3's comment "However, I am not entirely convinced of the overall benefits of employing this system if there exists a battery of well-characterized stable cell lines with tandem fluorescence tagged ATG8s. The main benefit would come from tagging cargo which would make the assay independent of the ATG8s (such as the authors have done with Halo-ER markers)."

2) Reviewer 1 "Does this assay can be used to measure autophagy influx in vivo? Authors may want to discuss this possibility in the discussion."

3) Line 188, Figure 2E: In order to convincingly conclude that there are no autolysosomes, can Figure 2e (especially the Baf treatment) be conducted using LAMP staining since the structures could represent autolysosomes that have lost their pH and therefore are not stained by lysotracker.

4) Figure 3G: There is a major band at approximately 37 kDa which also has in-gel fluorescence. Does this band represent free GFP or is it fluorescence from a cleavage product of the HaloTag construct? In addition, is the product localised to mitochondria or the cytosol (can be tested by crude mitochondrial isolation)? Western blotting with the GFP antibody would help clarify the identity of the species while gel fluorescence without TMR ligand could also be used. This is an important point to address because it could influence the interpretation of fluorescent signals observed via imaging.

*Reviewer #2 (Recommendations for the authors):*

Suggestions

1. Line 188, Figure 2E: In order to convincingly conclude that there are no autolysosomes, can Figure 2e (especially the Baf treatment) be conducted using LAMP staining since the structures could represent autolysosomes that have lost their pH and therefore are not stained by lysotracker.

2. Is the Halo-LC3B reporter influenced at all by alternative forms of lipidation including onto lysosomal membranes? For example, cGAMP treatment or C8 treatment -e.g. Goodwin et al. 2021 Science Advances? This would be helpful to know since the Halo-LC3B reporter might also report on non-autophagic events depending on the cellular context.

3. Figure 3G: There is a major band at approximately 37 kDa which also has in-gel fluorescence. Does this band represent free GFP or is it fluorescence from a cleavage product of the HaloTag construct? In addition, is the product localised to mitochondria or the cytosol (can be tested by crude mitochondrial isolation)? Western blotting with the GFP antibody would help clarify the identity of the species while in gel fluorescence without TMR ligand could also be used. This is an important point to address because it could influence the interpretation of fluorescent signals observed via imaging.

*Reviewer #3 (Recommendations for the authors):*

The data presented support the novel adaptation of the Halo-tag system and do recapitulate a method of monitoring autophagy that is more in line with that done in yeast. In addition provides a pulse-chase manipulation. However, I am not entirely convinced of the overall benefits of employing this system if there exists a battery of well-characterized stable cell lines with tandem fluorescence tagged ATG8s. The main benefit would come from tagging cargo which would make the assay independent of the ATG8s (such as the authors have done with Halo-ER markers).

---

## [Author Response]

Essential revisions:1) Modify the text to address reviewer3's comment "However, I am not entirely convinced of the overall benefits of employing this system if there exists a battery of well-characterized stable cell lines with tandem fluorescence tagged ATG8s. The main benefit would come from tagging cargo which would make the assay independent of the ATG8s (such as the authors have done with Halo-ER markers)."

We agree that one of the main benefits of the HaloTag reporters is that it could be used independently of ATG8s. However, the most significant advantage of the HaloTag reporters, including those containing ATG8, over currently available with tandem fluorescence tagged ATG8s is that they can be used in a pulse-chase manner. To emphasize this benefit of the HaloTag reporters, we have rewritten the text at Lines 266–271.

2) Reviewer 1 "Does this assay can be used to measure autophagy influx in vivo? Authors may want to discuss this possibility in the discussion."

We have already commented on this point in our previous version. We have added some more information to read “Furthermore, Halo-based reporters have been successfully employed in animal models (*England et al., 2015*) such as mice (*Masch et al., 2018; Rivas-Pardo et al., 2020*) and zebrafish (*Wan et al., 2019*), which suggests that the processing assay could be an additional method for assessing in vivo autophagic flux, e.g., when comparing autophagic flux among tissues” (Lines 306–310).

3) Line 188, Figure 2E: In order to convincingly conclude that there are no autolysosomes, can Figure 2e (especially the Baf treatment) be conducted using LAMP staining since the structures could represent autolysosomes that have lost their pH and therefore are not stained by lysotracker.

Thank you for pointing this out. We did not mean that there were no autolysosomes after Baf treatment. What we wanted to show is that Baf treatment blocks the degradation of mGFP in “autolysosomes”. Thus, we have deleted the sentence “suppressed the formation of Halo^SF650+^mGFP^–^ autolysosomes completely” and changed the phrase “the accumulation of Halo^SF650+^mGFP^+^ autophagosomes” to “the accumulation of Halo^SF650^ and mGFP in autophagosomes and autolysosomes” (Line 189). To make it more convincing, we have included data with the lysosome marker LAMP1-mRuby3 in Figure 2—figure supplement 1d, which confirms that Halo^SF650+^mGFP^+^ (arrowheads) and Halo^SF650+^mGFP^-^ puncta (arrows) are mostly negative and positive for LAMP1, respectively, without Baf, and some of the Halo^SF650+^mGFP^+^ puncta colocalize with LAMP1 (representing autolysosomes) after Baf treatment. We have referenced it in the text (Lines 183–192).

4) Figure 3G: There is a major band at approximately 37 kDa which also has in-gel fluorescence. Does this band represent free GFP or is it fluorescence from a cleavage product of the HaloTag construct? In addition, is the product localised to mitochondria or the cytosol (can be tested by crude mitochondrial isolation)? Western blotting with the GFP antibody would help clarify the identity of the species while gel fluorescence without TMR ligand could also be used. This is an important point to address because it could influence the interpretation of fluorescent signals observed via imaging.

Imaging results of HeLa cells stably expressing pSu9-Halo-GFP labeled with SF650 showed negligible signal outside of OMP25-mRuby3 labeled mitochondria (Author response image 1), which indicate that the cleavage product is localized to mitochondria. We have explained this issue in the text (Lines 222-226).

**Author response image 1. sa2fig1:** HeLa cells stably expressing pSu9-Halo-GFP in growing medium with 200 nM SF650 ligand.

Reviewer #2 (Recommendations for the authors):Suggestions1. Line 188, Figure 2E: In order to convincingly conclude that there are no autolysosomes, can Figure 2e (especially the Baf treatment) be conducted using LAMP staining since the structures could represent autolysosomes that have lost their pH and therefore are not stained by lysotracker.

Please see our response to Essential revision #3.

2. Is the Halo-LC3B reporter influenced at all by alternative forms of lipidation including onto lysosomal membranes? For example, cGAMP treatment or C8 treatment -e.g. Goodwin et al. 2021 Science Advances? This would be helpful to know since the Halo-LC3B reporter might also report on non-autophagic events depending on the cellular context.

Although we did not test its processing in situations when alternative forms of autophagy would be activated, we do not expect the Halo-LC3B reporter to be influenced by alternative forms of lipidation in the same way that LC3/Atg8 family-based probes would be. We have included a comment on this in the *Discussion section* to inform the readers (Lines 291-294).

3. Figure 3G: There is a major band at approximately 37 kDa which also has in-gel fluorescence. Does this band represent free GFP or is it fluorescence from a cleavage product of the HaloTag construct? In addition, is the product localised to mitochondria or the cytosol (can be tested by crude mitochondrial isolation)? Western blotting with the GFP antibody would help clarify the identity of the species while in gel fluorescence without TMR ligand could also be used. This is an important point to address because it could influence the interpretation of fluorescent signals observed via imaging.

Please see our response to Essential revision #4.

Reviewer #3 (Recommendations for the authors):The data presented support the novel adaptation of the Halo-tag system and do recapitulate a method of monitoring autophagy that is more in line with that done in yeast. In addition provides a pulse-chase manipulation. However, I am not entirely convinced of the overall benefits of employing this system if there exists a battery of well-characterized stable cell lines with tandem fluorescence tagged ATG8s. The main benefit would come from tagging cargo which would make the assay independent of the ATG8s (such as the authors have done with Halo-ER markers).

Please see our response to Essential revision #1.